# *Brassica juncea* L. (Mustard) *Extract Silver* NanoParticles and Knocking off Oxidative Stress, ProInflammatory Cytokine and Reverse DNA Genotoxicity

**DOI:** 10.3390/biom10121650

**Published:** 2020-12-09

**Authors:** Sohair Aly Hassan, Ali Mohamed El Hagrassi, Olfat Hammam, Abdelmohsen M. Soliman, Essam Ezzeldin, Wessam Magdi Aziz

**Affiliations:** 1Therapeutic Chemistry Department, Pharmaceutical Industries Division, National Research Centre, 33 El Bohouth St., Dokki, Giza 12622, Egypt; solimanmohsen@yahoo.com (A.M.S.); wessamagdi@yahoo.com (W.M.A.); 2Phytochemistry and Plant Systematics Department, Pharmaceutical Industries Division, National Research Centre, 33 El Bohouth St., Dokki, Giza 12622, Egypt; alielhagrasi@gmail.com; 3Pathology Department, Theodor Bilharz Research Institute, P.O. Box 30, El Warraq, Giza Governorate 12411, Egypt; totoali1@hotmail.com; 4Department of Pharmaceutical Chemistry, College of Pharmacy, King Saud University, P.O. Box 2457, Riyadh 11451, Saudi Arabia; esali@ksu.edu.sa

**Keywords:** Thioacetamide (TAA), acute liver toxicity, *Brassica juncea* L. seeds extract, LC/MS, phenolic compounds, pro-inflammatory cytokine, DNA genotoxicity

## Abstract

**Detoxification** is one of the main vital tasks performed by the liver. The purpose of this study was to investigate whether mustard in its normal or nanoparticles could confer a protective/therapeutic effect against TAA-induced acute liver failure in experimental animal models. Mustard ethanolic extract was analyzed by HPLC/MS. To induce liver failure, male rats were injected with 350 mg/kg bw TAA IP, then treated orally with a dose of 100 mg/kg for 15 d of mustard extract and its nanoform before and following induction. The levels of serum liver functions, total cholesterol (TCHo), total glyceride (TG), total bilirubin (TBIL), hepatic malonaldhyde (MDA) and nitric oxide (NO),glutathione (GSH), sodium oxide dismutase (SOD), as well as tumor necrosis factor (TNF-α,) and interleukin 6 (IL-6), were estimated. DNA genotoxicity and hepatic pathology, and immunohistologic (IHC) changes were assayed. The antioxidant content of Phenolic acids, flavonoids in mustard ethanolic extract substantially decreased the levels of ALT, AST, ALP and rehabilitated the histopathological alterations. In addition, nanoforms of mustard ethanol extract have notably increased the levels of GSH, SOD and significantly reduced the levels of MDA. The expression levels of TNF-α and IL-6 in serum and tissue were markedly downregulated. DNA genotoxicity was significantly reversed. Mustard introduced a protective and medicinal effect against TAA in both its forms.

## 1. Introduction

The liver is one of the most important organs responsible for the process of metabolism and selective uptake and clearance of drugs xenobiotic and environmental toxins, the matter which makes it very sensitive to drug toxicity [1]. Thioacetamide C2H5NS/TAA is an organosulfur compound known to induce acute or chronic liver disease (fibrosis and cirrhosis) in the experimental animal model very similar to that happen in humans. TAA toxicity results from a series of biotransformation into toxic metabolites that takes place in cytochrome P-450 enzymes in the liver [2].

TAA toxicity is triggered by its reactive metabolite sulfene or sulfone via the CYP450 system [3], which affects the most internal vital activities and causes oxidative stress and glutathione (GSH) depletion. Usually, in normal circumstances, these metabolites are detoxified by conjugating with antioxidative defense system (glutathione), but TAA toxic metabolites overwhelm the detoxification process and lead to a change in cell membrane permeability, disturb mitochondrial activity leading to lipid peroxidation, which contributes to the release of, cytokines, and prostaglandins [4,5] Thus, in initiating severe liver toxicity, the reactive oxygen species (ROS) may be the maestro by secreting a range of pro-inflammatory factors that could hurt DNA [6]. Considering the unappreciated side effects of chemically modified agents and the limited capacity of the modified pharmaceutical product to regulate major diseases, it is important to recognize new medicinal structures from other naturally occurring sources, including plant kingdom sources [7].

*Brassica juncea* L. seeds. Family *Brassicaceae*, is popularly known as Indian mustard, has both therapeutic and edible qualities. It has crucial contents of polyphenolic and phenolic compounds). Mustard preparations are well known for their mildly laxative, diuretic, and calming effect on the liver bile [8]. Mustard seeds since ancient times have been used by mankind for its culinary, as well as medicinal, properties. It has been systematically described and used in the classical Ayurvedic to purge the toxins out of the body [9]. The leaf extracts of *B juncea* have been reported to exhibit antioxidant, antinociceptive, and antihyperglycemic activities both in vitro and in vivo [10]. The leaf extracts of *B. juncea* have also been reported to significantly prevent the development of insulin resistance in rats [11]. Joint pain, nausea, alleviation of cough and colds, reducing swelling, and cranial cleaning were aptly treated with its extracted oil. Mustard oil has also been used for wound healing and skin diseases [9]. Nevertheless, the mechanism responsible for the protective effect of *B. juncea* seed extract in the context of TAA-induced liver failure in rat models still remains unknown.

In this study, we hypothesized that the nanotransformation of various phenolic and other compounds in mustard extract using AgNO_3_ couldimprove its benefits for health.

Thus, the purpose of this study was to elucidate the therapeutic and restorative impacts of mustard and its silver nanoparticles against TAA induced acute liver failure in animal models through speculating the biochemical parameters, assess the oxidative stress, anti-inflammatory impact and genotoxicity as well as histology and immunohistopathological studies.

## 2. Materials and Methods

### 2.1. Chemicals

All Chemicals, Silver nitrate and thioacetamide were procured from Sigma-Aldrich (St. Louis, MO, USA). The different solvents and chemicals used in the present study were of high analytical grade and supplied by a local vendor.

### 2.2. Authentication and Preparation of the Plant Extracts

Mustard seeds(*B. juncea*)were purchased from a local retail store(Cairo, Egypt). Mustard seeds were identified by the taxonomy Dept (Faculty of Science., Cairo University), where a voucher specimen was deposited for future reference). The seeds were cleaned, grounded with a coffee grinder to a fine powder. The powder was stored in a clean bottle at room temperature in a dark place. The powdered seed samples (100 g) were weighed and extracted with 300 mL of 70% ethanol. Then left for 72 h. for about 3 days using an end-to-end shaker at room temperature for 1 h. The extracts were filtered through Whatman no 1 filter paper and then dried by rotary evaporation. The yield of 12.5 g was kept at 4°C until use [12].

### 2.3. LC/MS Ytentative analysis for ethanolic Extract of B. juncea L.

Chemical and quantitative analysis of the main components of 100 g of mustard were analyzed. The sample (100 μg/mL) solution was prepared using high-performance liquid chromatography (HPLC) analytical grade solvent of/MeOH, filtered using a membrane disc filter (0.2 μm), then subjected to LC–ESI–MS analysis. Samples injection volumes (10 μL) were injected into the UPLC instrument equipped with reverse phase C-18 column (ACQUITY UPLC-BEH C18 1.7 µm particle size-2.1 × 50 mm Column). Sample mobile phase was prepared by filtering using 0.2 μm filter membrane disc and degassed by sonication before injection. Mobile phase elution was made with the flow rate of 0.2 mL/min using mobile gradient phase comprising two eluents: eluent A is H_2_O acidified with 0.1% formic acid, and eluent B is MeOH acidified with 0.1% formic acid. Elution was performed using the above gradient. The parameters for analysis were carried out using negative ion mode as follows: source temperature 150°C, cone voltage 30 eV, capillary voltage 3 kV, desolvation temperature 440°C, cone gas flow 50 L/h, and desolvation gas flow 900 L/h. Mass spectra scanning was detected in the ESI negative ion mode between *m*/*z* 100–1000. The peaks and spectra were processed using the MassLynx 4.1 software and tentatively identified by comparing its retention time (R_t_) and mass spectrum with reported data [13,14,15].

### 2.4. Synthesis of Mustard Silver Nanoparticles

*B. juncea*/mustard seed extract was used to synthesize (mustard-AgNPs). According to Hassan et al. [16], silver nitrate (0.017 g) was added in 100 mL of double distilled water to prepare 1 mM silver nitrate solution. Then, 1 mL of ethanolic mustard extract was added to 50 mL of silver nitrate (1 mM) solution and incubated for 24 h in a dark chamber to minimize photo-activation of silver nitrate at room temperature under static conditions for nucleation of the silver nanoparticles. The reduction of elemental Ag to AgO was confirmed by the color change from colorless to brownish-yellow, indicating that ethanolic mustard extract was encapsulated into the silver nanoparticles in the form of pellets. The suspensions were subjected to centrifugation for the pellets to settle, which were then dried using a vacuum dryer.

### 2.5. Characterization of Synthesized Silver Nanoparticles

UV-vis spectral analysis was done by using Shimadzu visible spectrophotometer (UV-1800, Japan). The periodic scans of the optical absorbance between 300 and 700 nm with a double-beam UV-visible spectrophotometer (Carry 100 with tungsten halogen light sources) were performed at room temperature to investigate the reduction of silver ions by seed extract.

Field emission scanning electron microscopy (FESEM), the morphology, particle dispersion, and chemical composition of the prepared Nanostructures were investigated by (FESEM) (Quanta 450) equipped with EDS at accelerating voltage 30 kV and further confirmed by transmission electron microscopy (TEM) for internal surface morphology using a JEOL JEM 1200 transmission electron microscope (JEOL Ltd., Peabody, MA, USA) [17].

### 2.6. Animals

A total of 64Westar rats weighing 180–200 g were taken from the breeding laboratory of the National Research Center and kept in an animal care facility. Rats were held in reserve in polypropylene cages at room temperature with a 12 h light–dark cycle and relative humidity of 60% ± 1%. Animals were provided purified water ad libitum with standard laboratory rat chow. The study was approved by the animal ethics committee of the Institutional Review Board of National Research Center [No 20,061]. The animal protocol was designed to lessen pain or distress to the animals. TAA (Sigma-Aldrich Chemical, ST. Louis, MO, USA) was dissolved in physiological saline, and the appropriately selected dose of 350 mg/kg bw was injected intraperitoneal (i.p.) in 1 mL volume [3].

#### Experimental protocol

The rats were divided into 8 groups consisting of 8 rats each as follows:Group (G 1): Normal healthy cont. (3% Tween-80) and water, respectivelyGroup (G 2): Intoxicated group was treated intraperitoneal (i.p.) with 350 mg/kg bw of freshly prepared TAA in a single shot [3], 5% glucose solution was added to the drinking water to avoid dehydration 24 h post-injection.Groups (G3, G4): were given orally; the standard plant extract and its nano forms (NMs) in a dose of I mL [100 mg/kg] for two weeks as a benchmark/control group of plant extract [12].Prophylactic groups (G5, G6): administered a dosage of 1 mL of standard mustard extract and/or its nanoform for two weeks prior to i.p. TAA insulting 350 mg/kg bw in one shot, by then continued with the same dosage of 1 mL of both extracts for two weeks.Treatment groups (G7, G8): were given orally; the standard plant extract and its nanoforms in a dose of 1 mL as a treatment for two weeks after (i.p.) single injection with the same assigned dose of TAA. Twenty-four hours following the last drug administration, blood samples were withdrawn from the retro-orbital plexus of the rats under light ether anesthesia. Then, rats were sacrificed by cervical dislocation under the same anesthesia for the collection of liver samples. A weighed part of the liver of each animal was rapidly dissected out, washed and homogenized using phosphate-buffered saline (PBS, 50 mM potassium phosphate, pH 7.5) at 4°C to produce a 20% homogenate. Liver homogenates were kept at −80°C until analysis. Another part of liver tissue was kept in 10% formalin-saline for histopathological and immunohistopathological examinations.

### 2.7. Biochemical Analysis

The liver function indices in terms of serum alanine aminotransferase (ALT), aspartate aminotransferase (AST) and alkaline phosphatase (ALP), total bilirubin (TBIL) and total lipid (TP) were estimated in the serum by colorimetric method using bio diagnostic kits (Biogamma, Rome, Italy, Stanbio, Barleben, Germany). Furthermore, total cholesterol (TC), triglyceride (TG) was estimated in the serum sample by enzymatic Colorimetric Method using bio diagnostic kits (Biogamma, Stanbio). The oxidative stress markers, lipid peroxidation (LPO), glutathione (GSH), superoxide dismutase (SOD) and nitric oxide (NO), were evaluated in the liver tissue homogenates according to Buege and Aust [18], Moron et al. [19], Nishikimiet et al. [20],Moshage et al. [21],respectively. Liver total protein contents were estimated following Bradford [22]. The DNA fragmentation index was estimated via the comet assay technique as described by Masoomi Karimi et al. [23]. The pro-inflammatory cytokines TNF-α, IL-6 were assayed in the serum sample using commercially available ELISA kits (R&D, Minneapolis, MN, USA).

### 2.8. Histopathological Studies

A division of the liver was preserved in 10% buffer formalin for at least 24 h. and then embedded in paraffin according to the standard protocol. Sections were cut into 5 μm-thick portions, transferred onto glass slides, and stained with hematoxylin and eosin (HE), investigated microscopically for any histopathological and immunohistochemical changes as well [24]

#### Histopathological and Immunohistochemical Examinations

Five micrometer-thick sections of paraffinized liver stained with hematoxylin and eosin were examined histopathologically under Zeiss microscope (Carl Zeiss Microscopy GmbH 07,745 Jena, Germany) with ×200, ×400 magnification powers. For immunohistochemical examinations of IL6, TNF α were performed on liver sections cuts from the paraffin blocks with a commercially available anti-mouse IL6, TNF α antibodies (Santa Cruz Biotechnology, CA, USA) at the optimal working dilution of 1:100. Briefly, slides were sectioned at 4 μm onto positively charged slides (Super frost plus, Menzel-Glaser, Germany), and the slides were stained on an automated platform (Dako Autostainer Link 48). Heat-induced antigen retrieval was used for 30 min at 97 °C in the high-PH Envision™ FLEX Target Retrieval Solution, and the primary antibody was used at a dilution of 1 in 100. The percent of positively stained brown cytoplasm (IL6, TNF α) were examined in 10 microscopic fields (under Zeiss light microscopy at x400).

### 2.9. Statistical Analysis

Statistical analysis of all data was expressed as mean ± SD of 8 rats in each group by using one-way analysis of variance (ANOVA), the CoStat software computer program accompanied with least significance difference (LSD) between groups at a significant level at *p* < 0.05. The percentage of improvement in both the prophylactic group and the treatment one was calculated according to the equation.

$$\%\;of\;improvement=\frac{Mean\;of\;TAA\;intoxicated\;group-Mean\;of\;treated\;group\times \;100}{Mean\;of\;control}$$


## 3. Results

The current results revealed that the incubation of 50 mL of 1 mM AgNO_3_ with 1 mL of mustard extract for 24 h at “room temperature” led to the synthesis of AgNPs as indicated by the development of the brown color. The spectrophotometric analysis showed a maximum absorption picture of the developed brown color at 425 nm (Figure 1).

### 3.1. The Visual Observation of Color Changes of Silver Nano Mustard (B. juncea) Extract

Moreover, the physical stability of the mustard-nano silver structures was evaluated in terms of ξ-potential. The surface charge of all the silver nanostructures of *B. juncea* seeds was negative, extending from [−10.6–13.3 mV]. The SEM &TEM check demonstrated that AgNPs molecule size was in the nanostructure in the scope of 4.8–39.5 nm (Figure 2A,B).

LC/MS results tentatively manifested twenty-eight phenolic compounds in the ethanolic extract of mustard. (Table 1, Figure 3). Fourteen phenolic acids and its glycosides were detected and determined as follow compounds 1, 2, 3, 4 and 5 were eluted at retention time (R_t_) 13.30, 15.22, 16.51 and 18.11 min., which produced a molecular ion peak [M − H]^−^ at *m*/*z*, 179, 163, 223, 193 and 191 and identified as caffeic acid, *p*-coumaric acid, sinapic acid, ferulic acid and quinic acid, respectively.

Peaks 6 and 7 are identified as 3-caffeoylquinic acid and 3-feruloylquinic acid detected with [M − H]^−^ at *m*/*z* 353 and 367, which yielded fragment ions at *m*/*z* (191, 179, 135) and (191, 193, 149). Compounds 8, 9 and 10 were eluted at retention time (R_t_) 21.20, 21.44 and 22.31 min. detected with [M − H]^−^ at *m*/*z* 355, 341 and 385, which referring to ferulic acid and caffeic acid after losing of hexose unit; thus, compound 8 was identified as ferulic acid hexoside, compound 9 was identified as caffeic acid hexoside, and compound 10 was identified as sinapic acid hexoside. Peaks 11, 12 and 13 were identified as *p*-coumaroyl malic acid, feruloyl malic acid and sinapoyl malic acid with [M − H]^−^ at *m*/*z* 279, 309 and 339. Compound 14 was identified as rosmarinic acid at [M − H]^−^ at *m*/*z* 359.

The two flavanol aglycone compounds peaks 15 and 16 were identified as kaempferol and quercetin with [M − H]^−^ at *m*/*z* 285 and 301. In addition to twelve flavanol glycoside compounds were identified as kaempferol *O*-dihexoside (compound 17), which appeared with molecular ion peak [M − H]^−^ at *m*/*z* 609 with fragments at *m*/*z* (477 and 285) corresponding to the kaempferol as aglycone after losing of two hexose units. In addition, peak 18 was identified as kaempferol-O-hexoside, which produced a molecular ion peak at *m*/*z* 447, corresponding to the kaempferol as aglycone after losing of hexose unit.

Compounds 19 and 20 are identified as kaempferol-*O*-cafeoyl dihexoside and kaempferol-*O*-p-coumaroyl dihexoside corresponding to a molecular ion peak [M − H]^−^ at *m*/*z* 771 and 755 with fragments *m*/*z* (609, 285) and (609, 447 and 285).

Peak 21 was eluted at retention time at 29.71 min with molecular ion peak [M − H]^−^ at *m*/*z* 801, which gave fragments at *m*/*z* 609, 447 and 285 which corresponding to kaempferol-*O*-hydroxyferuloyl dihexoside.

Compound 22 and 23 were identified as quercetin-*O*-trihexoside and quercetin-*O*-dihexoside corresponding to molecular ion peak [M − H]^−^ at *m*/*z* 787 and 625 with fragments (625, 463, 301) and (463, 301), which corresponding to quercetin as aglycone after losing three hexose units, while compound 23 lost two hexose units and quercetin as aglycone moiety.

Kaempferol-*O*-feruloyl dihexoside (compound 24) appears with molecular peak [M − H]^−^ at *m*/*z* 785 with fragments *m*/*z* (609, 447 and 285). Compound 25 and 26 were identified as quercetin-3-*O*-rutinoside and quercetin-*O*-hexoside, with molecular peak [M − H]^−^ at *m*/*z* 609 and 463. Peak 27 at retention time 33.5 min showed the molecular ion peak [M − H]^−^ at *m*/*z* 917 with fragments 755, 609, 447, 285, which was identified as kaempferol-*O*-*p*-coumaroyl trihexoside.

Finally, Compound 28 at retention time 33.80 min was observed with molecular peak [M − H]^−^ at *m*/*z* 963 with fragments as 801, 625 and 301, corresponding to quercetin-*O*-feruloyl trihexoside [13,14,15].

### 3.2. Effect of Brassica juncea L. (Mustard) Extract and Its Nanoforms on Liver Functions

The examination of ALT, AST and ALP are given in (Table 2). Single injections of TAA (350 mg/kg bw) significantly elevated the serum activities of ALT, AST and ALP by 2.3, 2.7 and 1.9-fold respectively at *P* <0.0001 compared to the normal control group. Administration of ethanolic extraction of *B. Juncea* at doses of 100 mg/kg bw was a prophylactic and/or treatment substantially dwindled the elevation of transaminases with different percentages of improvement, which are represented in Table 2.

### 3.3. Effect of Brassica juncea L. (Mustard) Extract in Their Normal and Its Nanoforms on Lipid Profile Parameters

Table 3 shows that experimental groups treated with TAA demonstrated a significant increase in the serum levels of total cholesterol (TC), triglycerides (TG), total lipids (TP), total bilirubin (TBIL) as compared with the healthy control group at (*p* < 0.05). Pretreatment with 100 mg/kg bw of *B. juncea* plant extract before and after intoxication with TAA and post-treatment with the same dose both resulted in a decrease of the above-mentioned parameters with mean levels differing from the normal and TAA groups at (*p* < 0.0001). However, the largest percentage of improvement change was achieved by nano plant extract rather than the remaining treatment classes.

### 3.4. Effect of Brassica juncea L. (Mustard) Extract in Their Normal and Its Nanoforms on Oxidative Stress Markers and Inflammatory Cytokines

A substantial decrease in the superoxide dismutase and glutathione levels in the challenged group following TAA injury p (*p* < 0.0001). However, pre- and post-treatment with *B. juncea* extract could improve the deterioration in the antioxidant parameters compared to the control group. Groups treated with mustard extract and its nano form exhibited the best improvement percentage rather than the pretreatment groups. In contrast with oxidative parameters, which showed an increase in each lipid peroxidation and nitric oxide levels in the challenged TAA intoxicated group at *p* < 0.0001 (Table 4). The treatment with Mustard extract (*B. juncea*) and its nanoforms showed a prominent reduction in oxidative parameters either as a prophylactic or as a treatment with different percentages of improvement (Table 4). The TAA-intoxication resulted in a remarkable elevation at (*p* < 0.05) in the measured inflammatory markers IL6 and TNFα by a 2.5-fold increase compared to the control group (Table 4). While *B. Juncea* extracts substantially reversed IL6 and TNFα levels in both pre- and post-treatment classes. In most antioxidant parameters as well as in inflammatory markers, the treatment with nanoform (TNMs) of *B. juncea* obtained the most amelioration outcomes.

### 3.5. Effects of Brassica juncea L. (Mustard) Extract and Its Nanoparticles on DNA Degradation

Fragmentation of DNA was assessed using the comet assay. Compared to the normal control group, TAA has substantially caused DNA damage. This DNA damage was demonstrated significantly in terms of an increase in the comet parameters presented as tail length with different three classes (Figure 4A,B), tail and tail moment DNA (percentage); the primary predictive parameter for DNA damage (Table 5). In the hepatotoxic community, there was a large broom-like tail and a small comet eye (Figure 4A), a typical component of toxic DNA damage and apoptosis. The length of the tail increases as DNA damage increases, as seen in (Figure 4A,B). In the meantime, administration of Mustard (*B. juncea*) both in its natural or in its nano pretreatment has counterbalanced the TAA effect as demonstrated by a considerable reduction in the length of the tail; this changed into extra glaring in the mustard nano form (Table 5).

### 3.6. Hematoxylin and Eosin Pathological Examination

Liver sections of normal rats showed normal cellular architectures, normal morphology, hepatocyte and central vein (Figure 5A). The TAA-intoxicated group displayed partial distortion of hepatic lobular architecture, with moderate hydropic degeneration of hepatocytes and binucleated hepatocytes accompanied by vein congestion with thick fibrous tissue in the portal tract (Figure 5B). Mustard (*B. juncea*) treatment (100 mg/kg) pre-post either in normal or nanoforms has shown observable healing effect presented, as shown in (Figure 5C–H).

### 3.7. Impact of Mustard Extract and Its Nanoform on the Inflammatory Markers through Immunohistochemistry

The anti-inflammatory effect of Mustard extract and its nano form was further evaluated via follow-up of the expression of IL-6 (Figure 6) and TNF-α (Figure 7) in hepatic tissues immunohistochemically. TAA intoxication 350 mg/kg bw resulted in a substantial elevation in TNF-α and IL-6 expression that were located mostly in the cytoplasm of hepatocytes in portal tract and fibrous septa compared to normal control group A, B. However, treatment with Mustard extract and its nanoforms either pre or after TAA intoxication improved and downregulated the expression of both TNF- α and IL6 Figure 6 and Figure 7 from (C–H), respectively, with the notion indicated that nano mustard extract has more potential to inhibit the inflammatory response induced by TAA intoxication.

## 4. Discussion

In the current study, green chemistry was employed to synthesize AgNPs using mustard seed extract (MSE). The change in the color of AgNO_3_ after 24 h incubation with (MSE) in the dark pointed to the ability of the extract of (MSE) to reduce AgNO_3_ to AgNPs. This result has been observed previously and indicated that the change in color might be appeared due to the surface plasmon resonance of the deposited AgNPs [25,26]. Formation of AgNPs was confirmed by the SEM and TEM as they ensured that the synthesized nanoparticles were within the nanoscale size 4.8–40 nm. Mustard is an ancient plant known since Biblical times; subsequent scientific and pharmacological studies verified its antimicrobial, analgesic, anti-inflammatory, antipyretic, antidiabetic and hepatoprotective qualities [2,27]. Perhaps and to the best of our knowledge, no more published data about the impact of mustard in acute liver failure.

The present results revealed through the LC/MS analysis that mustard (*B. juncea* L.) seeds extract (MSE) is rich in phenolic acids and flavonoid compounds, kaempferols glycosides, fatty acids. Many studies indicated that phenolic and flavonoids compounds are directly contributed to the antioxidant activities [28]. These antioxidant properties of the plant phenolic compounds are primarily due to the redox properties of these compounds, which enable them to donate hydrogen, act as singlet oxygen quenchers and reducing agents, as mentioned previously [29].

Serum liver enzymes are biomarkers to monitor and display the injure and aids in the clinical diagnosis of liver toxicity conditions [30]. When there is an injury to the liver due to any cause, then these enzymes are spilled into the blood move. This was in complete agreement with our results, represented by a pronounced increase in the **liver** enzymes AST, ALT, and ALP in the serum as a response to TAA injection. A leakage was coupled with a loss of liver architecture, hydropic degeneration, central vein obstruction and hepatic lymphocytes infiltration (Table 1, Figure 5). As thioacetamide is a potent centrilobular hepatotoxicant, its effects have been extended to increase cholesterol, glyceride, total lipid, total bilirubin levels indicating oxidative damage of the liver (Table 3). However, a substantive improvement and reduction in these parameters were observed after remedy with *B. juncea* in its both forms either as a prophylactic or as a treatment; however, the nanoforms have shown a greater significant effect on the elevated serum liver enzymes and the rehabilitation of the liver architecture as well. These outcomes go in line with Yang et al. [31] and seem to be consistent with what recently announced by Khaled [32], Melrose [27] and Le et al. [33] as they archived the amazing cell reinforcement and hepatoprotective properties of mustard against poisonousness instigated free radicals harm in the liver and furthermore with Walia et al. [11] as he detailed ethanolic leaf concentrate of *B. juncea* could be a superior medication of decision as a hepatoprotective plant source. The free radicals scavenging effects of mustard extracts are most in all likelihood attributed to its better polyphenols and flavones contents.

It is worth noting that the antioxidant impact of polyphenols is mainly due to their redox properties, which allow them to behave as reducing agents, hydrogen donors, singlet oxygen quenchers, metal chelators [34,35]. Moreover, our findings showed a marked decrease with a sensible percentage of improvements in the levels of cholesterol, total glycerides, total lipids and total bilirubin when treated with mustard extract in its normal and in its nanoform either pre-or post-TAA administration. Result consequences can be interpreted on the basis of the phytocomponents structure of mustard seeds extract to have the potential to act in a synergistic way to lessen hyperlipidemia and hypercholesterolemia and go with full coincidence with Lee et al. [36] when they used *B. juncea* L. leaf extract on fat deposition and lipid profiles. The current observations suggest *B. juncea* L. seeds be clinically functional for treating dyslipidemia, although the mechanisms underlying still needed to be investigated.

In another hand bile, acid synthesis plays a major role in hepatic regulation of cholesterol homeostasis as well as in the catabolism of cholesterol bilirubin is considered a physiologically important antioxidant [37], with beneficial effects at mildly elevated concentrations, and can neutralize ROS and prevent oxidative damage [38,39,40]. These results may also provide an explanation for the increment of total bilirubin as a justification mechanism of the body against TAA free radicals. Interestingly mustard extract in its two forms, either pre- or post-treatment dose, showed a significant improvement in bilirubin level. That improvement was pronounced with the nano mustard treatment formula to be in concurrence with Rawat et al. [41].

As unquestionably demonstrated, oxidative stress and inflammation are the two essential players driving forces behind various diseases. This was confirmed by the significant decrease in GSH and SOD levels along with the elevation in MDA, serum NO together with the marked disturbances in TNF-α and IL6 levels (Table 4). These outcomes may confirm the hepatotoxic TAA and its capability to cause hepatic damage and act as an initiator for a series of serious inflammatory changes via ROS generation and antioxidant protection mechanism regulation [42,43].

It is well established that plant extracts with antioxidant potential can protect against the oxidative damage caused by TAA hepatotoxicity [2]. The present study confirmed this observation and demonstrated that mustard extract and its nanoform substantially increased the levels of GSH and SOD, whether pre- or post-TAA injection. Our findings may be in full settlement with Keshari et al. [44] as they reported that stimulation of the Nrf2/ARE pathway is essential for antioxidant defense enzyme induction and intracellular GSH modulation in response to stress. Consequently, mustard can play an antioxidant role in acute liver damage. Interestingly, mustard extract and its nano remodulated the level of lipid peroxide, restoring it back to near normal. It additionally decreased the overproduction of NO, which is commonly linked to the lack of cytochrome P450 content and injury in rat hepatocytes [45].

Moreover, a marked increase in serum levels of TNF-α and IL6 pro-inflammatory markers had been recorded in the challenged TAA group in both serum and tissue in comparison to the control results (Table 4 and Figure 6). It can clearly be seen the expression of TNF-αandIL6 is about 70%. These findings may ensure that the entire inflammatory cascade, including the activation of transcription factors, is activated when the redox status is attached [46,47,48]. However, these parameters were remodulated and downregulated by mustard extract treatment in both normal and its nano form either pre-or post TAA intoxication. These good responses may be supportive of Le et al. [33] as they suggested the anti-inflammatory properties of the *B. juncea* spice/herb (Figure 6).

From a near perspective, data from this research showed that thioacetamide could interact with DNA and induce changes in the hepatic cells (Figure 4). While TAA/challenged group damage DNA was demonstrated by an increase in the tail moment. However, *B. juncea* therapy. Ethanolic extract in normal or nanoform decreased DNA damage dramatically and reversed the comet’s length back to normal. These findings can be interpreted on the basis of the antioxidant phytochemical components contained in the mustard extract, which carry a high flavonoid diffusion rate into the membranes, allowing them to scavenge oxy radicals in the lipid bilayer at several sites. The results may be in harmony with the results of Zargar et al. [49], as he used rutin before two weeks of TAA assault and resulted in the complete reversal of TAA-mediated hepatic toxicity. Our results also go in the same connection with Ansar et al. [50], who proposed that flavonoids may also be involved in the indirect induction of detoxifying genes, which may promote detoxification of TAA and decrease its toxicity.

It is noteworthy that this study was designed to investigate the mustard extract in its natural or in its nano sort either as a prophylactic or as a treatment against the insult of TAA inducing acute liver failure. We would like to draw to the reader’s attention that treatment with mustard nano form has shown superiority over the ethanol normal extract formula. The matter which reflects the efficiency of the nanoparticles in term of their existence of the essential functional groups to make AgNPs very stable and prevent it from being anchored. In addition, the existence of the bioactive molecules on the surface of AgNPs may increase its efficacy and bioavailability. It can also enhance its solubility, improve plasma half-life, preventing degradation in the intestinal environment and increasing permeation in the small intestine, which may confer strong antioxidative activity [51,52]. Although the results of the present study support mustard as a great candidate for liver, It appears that mustard and its nanoparticles may initiate comets, Table 5. Mustard in both forms cannot efficiently switch/ reverse the DNA fragmentations back or close to the normal. This result may be credited to a few leftover pesticides within the mustard seed itself or may predict some toxic components contained within the mustard. Azubuike et al. [12]. Duy and Trang [53] recorded that *B. juncea* seeds contain saponins, flavonoids and tannins. In expansion, mustard oil is well known to contain glycerides of erucic acid, which is considered harmful to human wellbeing. Hence, these observations recommend that drawn-out utilization of *B. juncea* seed extract may have some hepatotoxic impacts. The matter which motivates more extensive research in order to explore the behind plausible mechanism.

## 5. Conclusions

Our experimental findings demonstrated that the plant under investigation (mustard) possesses a potent antioxidative-inflammatory effect, and its nanostructure is more potent than its typical. The LC/MS Phytochemical analysis of the plant extract demonstrates the presence of a plethora of powerful compounds like phenolic and flavonoids compounds, which brought possible mechanism of hepatoprotective activity due to their free radical scavenging and antioxidant properties in addition to their anti-inflammatory effects. These properties are mediated by downregulating the liver function enzymes, lipid profile, oxidative stress, expression of TNF-α, and IL-6.in both serum and tissue. Although we acknowledge these results as not only has it been revealed that the ethanolic concentration of *Brassica juncea* L. has a reasonable measure of safety and risks, the study also provided new insights into the possible role of mustard in the treatment of acute liver injury.

## Figures and Tables

**Figure 1 biomolecules-10-01650-f001:**
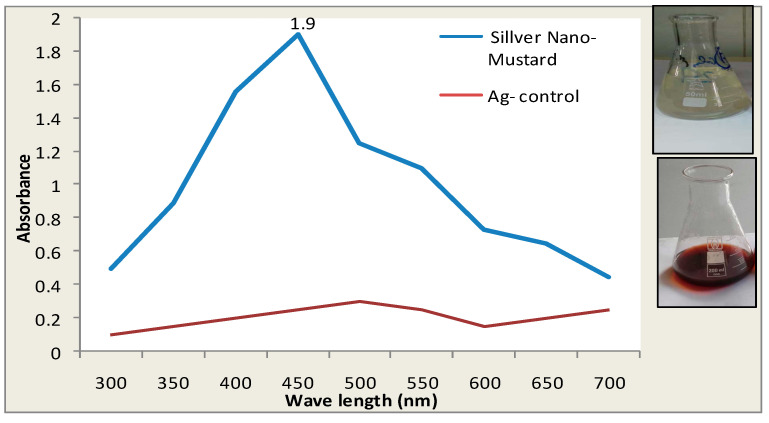
UV-visible spectrophotometric analysis of Ag mustard nanoparticles. Maximum absorption of Ag mustard nanoparticles at a wavelength of 425 nm.

**Figure 2 biomolecules-10-01650-f002:**
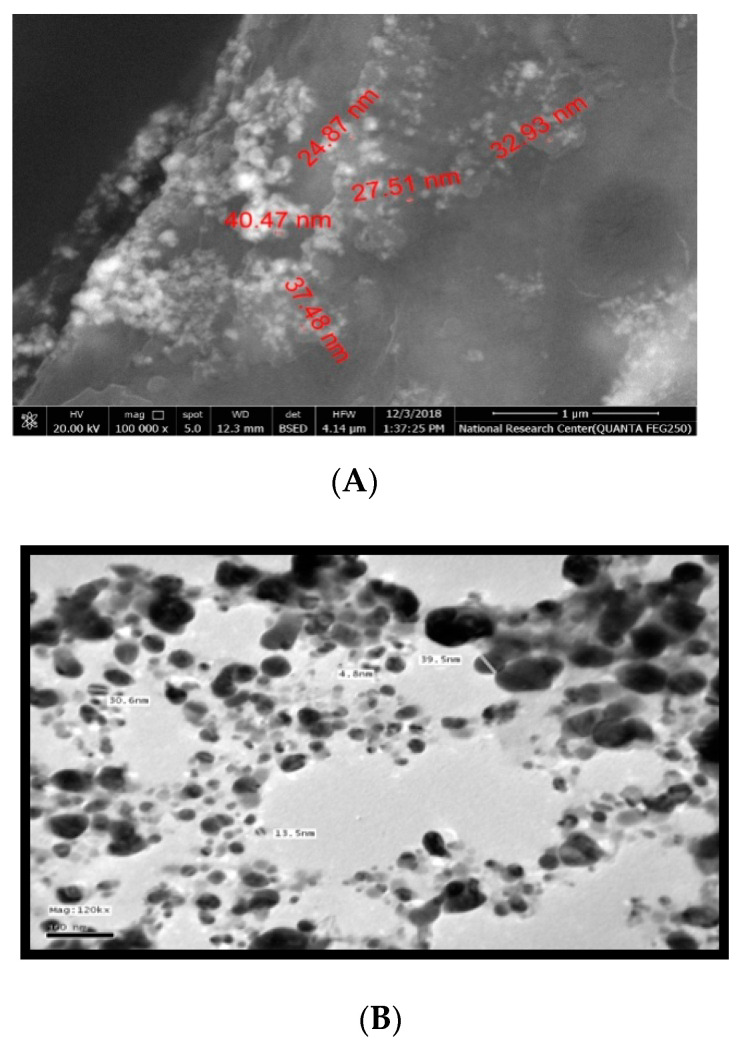
SEM (**A**) and TEM (**B**) micrographs of round state of Ag-nanoparticles of mustard with a size range from 4.8 to 39.5 nm.

**Figure 3 biomolecules-10-01650-f003:**
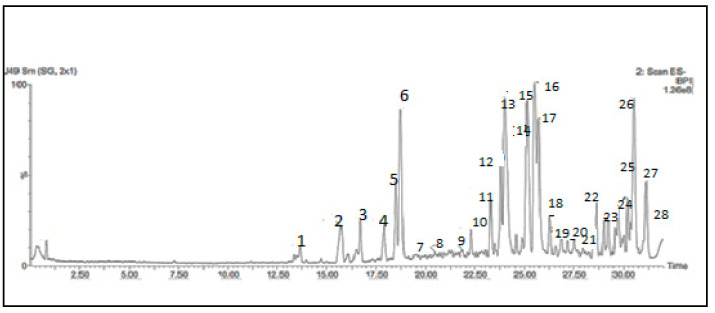
Total ion chromatogram of LC–MS of ethanolic extract of MS.

**Figure 4 biomolecules-10-01650-f004:**
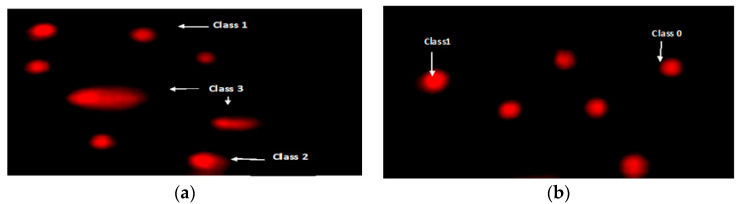
Comet assay in liver tissues shows visual DNA damage score: (**a**) classes 1, 2 and 3, (**b**) class 0 and 1.

**Figure 5 biomolecules-10-01650-f005:**
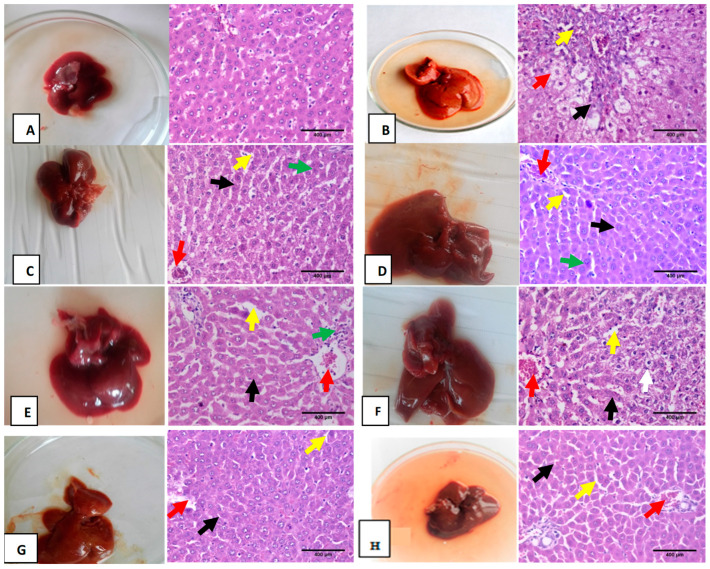
Photomicrograph of sections of hematoxylin and eosin (H and E, ×400) for liver tissues of different treated groups. (**A**) represents the sections from the normal control group with typical structure and morphology of the liver tissue; (**B**) TTA-intoxicated group showed severe abnormalities, partial distortion of lobular hepatic architecture (black arrow), with moderate hydropic degeneration of hepatocyte (red arrow) and thick fibrous tissue in portal tract (blue arrow); (**C**,**D**) liver sections treated with MS and nano MS showed hepatic tissue with almost normal intact hepatic lobular architecture and structure, hepatocytes arranged in thin plates (black arrow) and sinusoids (yellow arrow), central vein (red arrow), binucleated nuclei (green arrow); (**E**,**F**) liver sections treated orally with PMS nano PMS as a prophylactic group. They demonstrated tangible improvement as hepatic tissue was almost in normal intact hepatic lobular architecture and structure; hepatocytes arranged in thin plates (black arrow) with mild interlobular inflammatory infiltration (green arrow) and sinusoids (yellow arrow), congested central vein (red arrow), hepatocytes with moderate hydropic degeneration (white arrow); (**G**,**H**) liver sections treated with MS and MS nano groups showed improvement in hepatic tissues as it appeared in intact lobular hepatic architecture, hepatocyte with thin plates (black arrow), congested central vein (red arrow) and congested sinusoids (yellow arrow) ((**H**,**E**), ×400).

**Figure 6 biomolecules-10-01650-f006:**
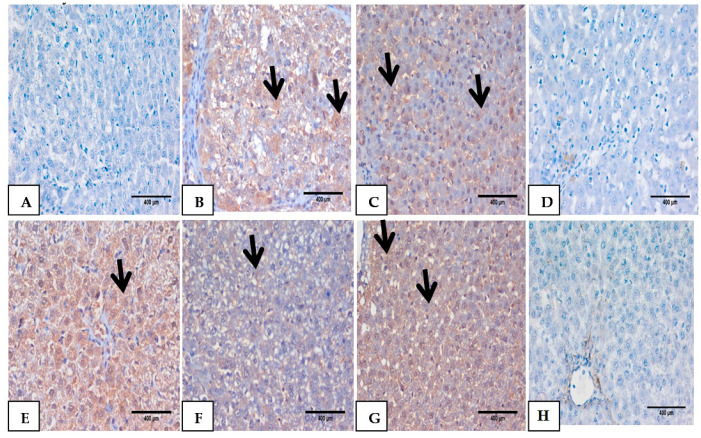
(**A**) Liver section stained with immunohistochemistry for IL6, (**B**) represents TAA +Ve intoxicated control group as it showed a dense expression of IL6 as a brownish cytoplasmic stain of hepatocytes in about 60% (black arrow), (**C**,**D**) liver sections pretreated with Ms and NMs showed a very light mild density of IL6 as brownish cytoplasmic stain 20%,while NMs group sections showed a pronounced effect with negative expression of IL6 (black arrow), (**E**,**F**) liver sections from prophylactic (Ms and NMs) showed moderate improvement in brownish cytoplasmic stain IL6 expression 10–50% (black arrow), (**G**,**H**) Liver sections treated with Ms and NMs show a marked improvement with a very light expression of IL6 10–50% as a brownish cytoplasmic stain (black arrow) (DAB,IHC, ×400).

**Figure 7 biomolecules-10-01650-f007:**
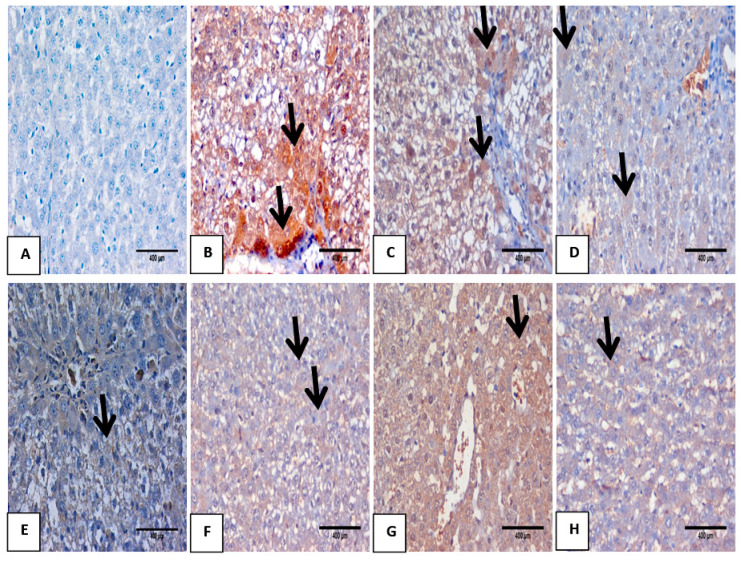
Liver section stained with immunohistochemistry for TNF-α.from(**A**) normal control group showed negative expression of TNF-α; (**B**) TAA intoxicated liver (positive control) showed marked expression of TNF-α as a brownish cytoplasmic stain of hepatocytes in about ± 70% (black arrow); (**C**,**D**) oral treatment with Ms and NM. As +ve control for mustard showed very mild expression of TNF-α as a brownish cytoplasmic stain of hepatocytes in about 10–20% (black arrow); (**E**,**F**) liver section from pretreated groups of Ms and NMs demonstrated moderate expression of TNF-α as a brownish cytoplasmic stain of hepatocytes in about ± 20–30% (black arrow); (**G**,**H**) liver sections treated with Ms and NMs showed marked improvement of TNF-α expression as a brownish stain of hepatocytes in about 25–50% (black arrow) (DAB, IHC, ×400).

**Table 1 biomolecules-10-01650-t001:** LC–MS analysis of ethanolic *Brassica juncea* L. (mustard) extract.

Peak No	R_t_ (min.)	[M]	[M − H]^−^	Fragments *m*/*z*	Tentative Identification
1	13.3	180	179	135	Caffeic acid
2	15.22	164	163	119	*p*-Coumaric acid
3	16.51	224	223	179	Sinapic acid
4	18.11	194	193	149	Ferulic acid
5	19.21	192	191	-----	Quinic acid
6	19.52	354	353	191,179,135	3-caffeoylqunic acid
7	20.1	368	367	191, 193, 149	3-feruloylquinic acid
8	21.2	356	355	193, 149	Ferulic acid hexoside
9	21.44	342	341	163, 135	Caffeic acid hexoside
10	22.31	386	385	203, 179	Sinapic acid hexoside
11	23.23	280	279	163, 133, 119	*p*-Coumaroyl malic acid
12	23.51	310	309	193, 149, 133	Feruloyl malic acid
13	24.12	340	339	223, 179, 133	Sinapoy lmalic acid
14	25.05	360	359	----------	Rosmarinic acid
15	25.55	287	286	----	Kaempferol
16	25.41	302	301	----	Quercetin
17	26.51	610	609	477, 285	Kaempferol-*O*-dihexoside
18	27.61	448	447	285	Kaempferol-*O*-hexoside
19	28.52	772	771	609, 285	Kaempferol-*O*-cafeoyl dihexoside
20	29.50	756	755	609, 447, 285	Kaempferol-*O*-*p*-coumaroyl dihexoside
21	29.71	802	801	609, 447, 285	Kaempferol-*O*-hydroxyferuloyl dihexoside
22	29.81	788	787	625, 463,301	Quercetin-*O*-trihexoside
23	30.10	626	625	463, 301	Quercetin-*O*-dihexoside
24	31.21	786	785	609, 447, 285	Kaempferol-*O*-feruloyl dihexoside
25	32.21	610	609	301	Quercetin-3-*O*-rutinoside
26	33.32	464	463	301	Quercetin-*O*-hexoside
27	33.50	918	917	755, 609, 447, 285	Kaempferol-*O*-p-coumaroyl trihexoside
28	33.80	964	963	801, 625, 301	Quercetin-*O*-feruloyl trihexoside

**Table 2 biomolecules-10-01650-t002:** Restorative and prophylactic impacts of *Brassica juncea* L. (mustard) extract in its normal and its nanostructure on liver function parameters.

Groups Parameters	ALT(U/mL) Mean ± SD	AST(U/mL) Mean ± SD	ALP(U/L) Mean ± SD
	Improvement%	Improvement%	Improvement%
Control (-ve)	53 ^b^ ± 7.4	56.2 ^b^ ± 8.07	115.4 ^d^ ± 14.70
TAA (+ve)	124 ^a^ ± 10.20	156.6 ^a^ ± 9.64	215.4 ^a^ ± 9.36
Control-Mustard (Ms)	51.4 ^b^ ± 8.45	61.2 ^b^ ± 6.05	138.2 ^c^ ± 6.18
Control- Ag nano Mustard(NMs)	62.8 ^b^ ± 7.89	68.8 ^b^ ± 8.28	125.4 ^cd^ ± 13.65
Prophylactic-Mustard(PMs)	68 ^b^ ± 8.60 (105%)	66.6 ^b^ ± 6.35 (163%)	171 ^b^ ± 12.74 (38%)
Prophylactic Ag-nano-Mustard(PNMs)	62.2 ^b^± 9.20 (116%)	62.8 ^b^ ± 8.05 (168%)	132.4 ^cd^ ± 7.04 (72%)
Treated-Mustard(TMs)	58.2 ^b^ ± 8.50 (125%)	54.6 ^b^ ± 7.75 (182%)	141.2 ^c^ ± 5.26 (64%)
Treated Ag nano-Mustard(TNMs)	55.4 ^b^ ± 11.09 (130%)	57.8 ^b^ ± 8.07 (177%)	121.40 ^cd^ ± 15.63 (82%)

Data are mean ± SD of 8 rats in each group. Statistical analysis was done using one-way analysis of variance (ANOVA) using the CoStat computer program accompanied by the least significant level (LSD) between groups at *p* < 0.05. Unshared superscript letters are significant values between groups at *p* < 0.0001.

**Table 3 biomolecules-10-01650-t003:** Restorative and prophylactic impacts of *Brassica juncea* L. (mustard) extract in its normal and its nanostructure on lipid profile.

Groups Parameters	Total Cholesterol (mg/dL) Mean± SD	Total Glycerides (mg/dL) Mean± SD	Total Lipids (g/dL) Mean± SD	Total Bilirubin (mg/dL) Mean± SD
	Improvement%	Improvement%	Improvement%	Improvement%
Control (−ve)	40.36 ^d^ ± 1.8	100.6 ^e^ ± 12.02	0.39 ^d^ ± 0.035	1.15 ^b^ ± 0.095
TAA (+ve)	90.25 ^a^ ± 2.37	283.2 ^a^ ± 18.99	0.92 ^a^ ± 0.036	2.34 ^a^ ± 0.125
Control-Mustard(Ms)	58.81 ^b^ ± 9.45	135.8 ^d^ ± 12.40	0.44 ^d^ ± 0.035	1.22 ^b^ ± 0.09
Control-Ag nano Mustard(NMs)	44.65 ^cd^ ± 3.51	103.6 ^e^ ± 12.70	0.42 ^d^ ± 0.025	1.17 ^b^ ± 0.065
Prophylactic-Mustard(PMs)	56.4 ^b^ ± 3.04 (85%)	196 ^b^ ± 23.60 (87%)	0.66 ^b^ ± 0.050 (67%)	1.25 ^b^ ± 0.11 (95%)
Prophylactic-Ag nano-Mustard(PNMs)	50.2 ^c^ ± 3.70 (99%)	166.8 ^c^ ± 16.60 (116%)	0.59 ^c^ ± 0.040 (85%)	1.36 ^b^ ± 0.095 (85%)
Treatment-Mustard(TMs)	60.5 ^b^ ± 3.65 (73%)	193.8 ^b^ ± 12.05 (89%)	0.71 ^b^ ± 0.052 (54%)	1.29 ^b^ ± 0.135 (91%)
Treatment-Ag nano-Mustard(TNMs)	57.52 ^b^ ± 4.085 (80%)	173.6 ^bc^ ± 16 (109%)	0.66 ^b^ ± 0.070 (67%)	1.30 ^b^ ± 0.11 (90%)

Data are mean ± SD of 8 rats in each group. Statistical analysis was done using one-way analysis of variance (ANOVA) using the CoStat computer program accompanied by the least significant level (LSD) between groups at *p* < 0.05. Unshared superscript letters with significant values between groups at *p* < 0.0001.

**Table 4 biomolecules-10-01650-t004:** Impacts of *Brassica juncea* L. (mustard) and its nanostructure on ROS and inflammatory cytokines.

Groups Parameters	SOD (Umol/mg Protein) Mean± SD	Glutathione (ug/mg Protein) Mean± SD	Lipid Peroxidation (Umol/mg Protein) Mean± SD	Nitric oxide (mmol/g Tissue) Mean± SD	IL6 (Pg/mL) Mean± SD	TNF-α (Pg/mL) Mean± SD
	Improvement%	Improvement%	Improvement%	Improvement%	Improvement%	Improvement%
Control (-ve)	125.21 ^a^ ± 8.60	25.90 ^a^ ± 3.1	3.03 ^c^ ± 0.17	11.28 ^d^ ± 1.70	98.87 ^f^ ± 2.35	62.38 ^f^ ± 1.71
TAA (+ve)	53.42 ^d^ ± 7.25	10.10 ^c^ ± 2.05	7.5 ^a^ ± 0.47	34.96 ^a^ ± 2.25	236.77 ^a^ ± 9.95	159.9 ^a^ ± 2.14
Control-Mustard (Ms)	98.45 ^bc^ ± 4.03	23.17 ^a^ ± 2.7	4.11 ^b^± 0.51	18.36 ^b^ ± 1.9	125.18 ^d^ ± 5.33	75.91 ^e^ ± 5.46
Control- Ag nano Mustard(NMs)	92.08 ^c^± 8.60	24.98 ^a^ ± 1.8	4.14 ^b^ ± 0.25	13.78 ^cd^ ± 2.24	108.83 ^e^ ± 5.18	73.48 ^e^ ± 4.61
Prophylactic-Mustard(PMs)	90.63 ^c^ ± 9.64 (29%)	16.72 ^b^ ± 1.65 (26%)	4.55 ^b^ ± 0.25 (97%)	18.90 ^b^ ± 2.50 (142%)	155.54 ^b^ ± 4.52 (82%)	139.86 ^b^ ± 5.15 (32%)
Prophylactic Ag nano-Mustard(PNMs)	94.08 ^c^ ± 7.35 (32%)	24.68 ^a^ ± 2.9 (56%)	3.89 ^b^ ± 0.30 (119%)	15.97 ^bc^ ± 1.70 (168%)	141.33 ^c^ ± 5.04 (97%)	127.61 ^c^ ± 5.14 (51%)
Treatment-Mustard(TMs)	105.7 ^bc^ ± 9.85 (42%)	26.28 ^a^ ± 2.85 (62%)	3.90 ^b^ ± 0.49 (118%)	18.35 ^b^ ± 2.70 (147%)	148.05 ^c^ ± 4.52 (90%)	135.65 ^b^ ± 3.85 (39%)
Treatment Ag nano-Mustard(TNMs)	111.09 ^b^ ±10.85 (46%)	25.61 ^a^ ± 3.9 (60%)	4.12 ^b^ ± 0.52 (112%)	16.43 ^bc^ ± 1.44 (164%)	120.38 ^d^ ± 6.54 (118%)	122.15 ^d^ ± 3.07 (60%)

Data are represented as mean ± SD of 8 rats in each group. One-way analysis of variance (ANOVA) was employed using the CoStat computer program accompanied by the least significant level (LSD) between groups at *p* < 0.05. Unshared superscript letters are significant values between groups at *p* < 0.0001.

**Table 5 biomolecules-10-01650-t005:** Effects of *Brassica juncea* L. (mustard) and its nanoform on DNA degradation.

Treatment	No. of Cells		Class ^¥^ of comet				DNA Damaged Cells (Mean ± SD)
	Analyzed	Total Comets	0	1	2	3	
Control negative	500	34	466	23	11	0	6.83 ± 0.11 ^d^
TAA Control positive	500	124	376	32	44	48	24.81 ± 0.82 ^a^
Ms	500	83	417	35	26	22	16.62 ± 0.46 ^b^
AgNMs	500	52	448	19	18	15	10.43 ± 0.41 ^c^
Prophylactic-mustard PMs	500	79	421	33	25	21	15.82 ± 0.61 ^b^
Prophylactic Ag nano-mustard PNMs	500	59	441	21	23	15	11.80 ± 0.44 ^c^
Treatment TMs	500	67	433	27	23	17	13.44 ± 0.36 ^bc^
Treatment T AgNMs	500	51	449	17	19	15	10.20 ± 0.28 ^c^

**^¥^**-Class 0 = no tail; 1 = tail length < diameter of nucleus; 2 = tail length between 1X and 2X the diameter of nucleus; and 3 = tail length >2X the diameter of nucleus.

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
