# Peer review of "Brassica juncea L. (Mustard) Extract Silver NanoParticles and Knocking off Oxidative Stress, ProInflammatory Cytokine and Reverse DNA Genotoxicity"

_biomolecules, 2020, doi:10.3390/biom10121650_

Round 1

Reviewer 1 Report

The authors investigated the effects of mustard and its silver nanoparticles on thioacetamide-induced liver failure in animal models. Their hypothesis is that the nanotransformation of various phenolic and other compounds in mustard extract using AgNCO3 could improve its benefits for health. For this purpose, they caused liver failure by intraperitoneal injection 350 mg/kg b.w of TAA in male treated with mustard and its nanoform 100 g/kg b.w before and after induction for15 days. They found that mustard in both forms significantly decreased levels of ALT, AST, ALP and rehabilitated histopathological changes. In addition, nanoforms of mustard ethanol extract increased the levels of GSH, SOD and significantly reduced the levels of MDA as a marker of lipid peroxidation. The expression levels of TNF-α and IL-6 in serum and tissue were decreased. DNA genotoxicity was also prevented. They concluded that mustard might have a protective and medicinal effect against TAA in both its forms. Many experimentations performed with significant results.

However I have the following comments and questions:

1) The main limitation of this paper is the wrong statistical test. The authors used a t-test which is incorrect. Since they have more than two parameters (TAA, mustard, ect), they have to use a two-way ANOVA.

2) Since the statistics is wrong, the interpretation of the all data is completely inappropriate.

3) The reading of this paper is difficult due to many grammatical errors in the text. These errors can be grouped into the use of inappropriate words. Thus, the manuscript has to benefit by a review by an editor.

4) How did they find the dose of TAA and mustard?

5) In tables 2, 3 and 4, what does mean “improvement in %”? Do the data indicate mean±SD or percentage? Please specify and correct.

6) In general, the authors should show graphs instead of Tables. It is easier for the reviewer and the potential future readers to understand these data. Please change them.

7) What is exactly the mechanism explaining the protection of mustard on liver TAA toxicity? Where does mustard exactly act on the hepatocytes? On mitochondria?

8) Is mustard an antioxidant? If yes, the authors have to measure the ROS content in liver and show the prevention with mustard. They have also to investigate the source of ROS? From mitochondria for instance?

9) What about apoptosis?

10) Table 3: the results are quite surprising. Does mustard act on the cholesterol biosynthesis? Please discuss this point.

11) The authors have to discuss the clinical translation of these findings.

Author Response

Dear Editors,:

Thank you very much for the review of our manuscript entitled: “Brassica juncea L(mustard) Nano Particles and Knocking off Oxidative Stress, Pro-Inflammatory Cytokine and Reverse DNA Genotoxicity “

Responses to the reviewer's comments

We would like to thank Reviewers for taking the time and effort necessary to review the manuscript. We sincerely appreciate all valuable comments and suggestions, which helped us to improve the quality of the manuscript. Our responses to the Reviewers’ comment are described below in a point-to-point manner. Appropriated changes, suggested by the Reviewers, have been introduced to the manuscript (highlighted within the document).

I would like to draw the reviewer attention that the line number has been changed due to the corrections done.

(Reviewer 1):

Comments and Suggestions for Authors

The authors investigated the effects of mustard and its silver nanoparticles on thioacetamide-induced liver failure in animal models. Their hypothesis is that the nanotransformation of various phenolic and other compounds in mustard extract using AgNCO3 could improve its benefits for health. For this purpose, they caused liver failure by intraperitoneal injection 350 mg/kg b.w of TAA in male treated with mustard and its nanoform 100 g/kg b.w before and after induction for15 days. They found that mustard in both forms significantly decreased levels of ALT, AST, ALP and rehabilitated histopathological changes. In addition, nanoforms of mustard ethanol extract increased the levels of GSH, SOD and significantly reduced the levels of MDA as a marker of lipid peroxidation. The expression levels of TNF-α and IL-6 in serum and tissue were decreased. DNA genotoxicity was also prevented. They concluded that mustard might have a protective and medicinal effect against TAA in both its forms. Many experimentations performed with significant results.

However I have the following comments and questions:

  • The main limitation of this paper is the wrong statistical test. The authors used a t-test which is incorrect. Since they have more than two parameters (TAA, mustard, ect), they have to use a two-way ANOVA.

Reply:  Kindly be informed that the main aim of the study is to study the effect of either mustard extract OR its nanoform (one factor at a time) on TAA-induced acute liver failure in the 8 experimental animal groups. So we meant to study the effect of the extract on liver functions , lipid profile  , oxidative stress markers and inflammatory cytokines in these groups before and after TAA toxicity as one  variant at a time .Then we compared the results of each variants separately between the 8 groups using one-way ANOVA followed by LSD to detect the group responsible for the significant result of the ANOVA if any exists.

we didn’t use two way ANOVA or a t-test we used one way ANOVA

One way ANOVA compare the means of different groups (more than two groups) by one variable form each parameters [ALT, AST,].

While two way NOVA compare the means of different groups by more than one variable from each parameters.

In our research we used one way ANOVA not two way ANOVA  or T-test as we compared each parameters from ALT, AST,ALP,…… as one independent variable with the different groups.

2) Since the statistics is wrong, the interpretation of the all data is completely inappropriate

Reply:  Depending on the above mentioned explanation for the statistical method, We hope to be more clearer, understandable and the interpretation is appropriate.

3) The reading of this paper is difficult due to many grammatical errors in the text. These errors can be grouped into the use of inappropriate words. Thus, the manuscript has to benefit by a review by an editor.

 Reply: The whole manuscript has reviewed and grammatically corrected’

4) How did they find the dose of TAA and mustard?

Reply: 350 mg/kg/bw of freshly prepared TAA was applied according to the previous studies reference ( 3) [ Koblihová, et al.,. 2014). The dose of Mustard was obtained from (Azubuike et al.,2019 )reference 12 in the main text.

5) In tables 2, 3 and 4, what does mean “improvement in %”? Do the data indicate mean±SD or percentage? Please specify and correct.

Reply: In addition to the mean ±SD we calculated the percentage of improvement happen in the treated groups in comparison to the intoxicated TAA positive control group.    We used the following equation

    Section -2.9

6) In general, the authors should show graphs instead of Tables. It is easier for the reviewer and the potential future readers to understand these data. Please change them.

Thank for your value advice, but as tables are more explanatory and contain more details we prefer to translate our statistical info in tables especially in case of presence of calculation of percentage of improvements for more groups

7) What is exactly the mechanism explaining the protection of mustard on liver TAA toxicity? Where does mustard exactly act on the hepatocytes? On mitochondria?

The effect of mustard on the hepatocyt is already shown on table 2,3,4 and supported by histological studies (Figure 5,6A,B ) , Mustard most probably acts on the mitochondria as it’s the main source of oxidative stress .  The suggested mechanism may be via Serum liver enzymes  which are considered as biomarkers to monitor and display the injure and aids in the clinical diagnosis of liver toxicity conditions, or via stimulation of the Nrf2/ARE pathway which is essential for antioxidant defense enzyme induction and intracellular GSH modulation in response to stress, all these interpretations are already explained in discussion paragraph line 406- 414 But the exact mechanism  by which mustard act on hepatocyte is not fully known yet.

8) Is mustard an antioxidant? If yes, the authors have to measure the ROS content in liver and show the prevention with mustard. They have also to investigate the source of ROS? From mitochondria for instance?

 Yes. Mustard is very rich source with polyphenols and flavones which are known to have a strong power of antioxidant . Moreover the LC-Mass analysis in the present study proved by evidence that mustard is very rich by antioxidant components Table -1. In addition many studies have declared, and confirmed the antioxidant properties of mustard which is rich in phenolic acids and Flavonoid compounds, Kaempferols glycosides [27] fatty acids. Many studies indicated that phenolic & flavonoids compounds are directly contributed to the antioxidant activities. [28] These antioxidant properties of the plant phenolic compounds are primarily due to redox properties of these compounds, which enable them to donate hydrogen, act as singlet oxygen quenchers and reducing agents as mentioned previously. [29].

8) 2nd part: They have also to investigate the source of ROS? From mitochondria for instance?    Reply :  YES The mitochondrion is a major source of reactive oxygen species  but it is addressed here through estimating some antioxidant parameters in liver tissue like SOD, NO, Lipid peroxide, table 4.

9) What about apoptosis? 

Reply: Apoptosis is already  assessed  via TNF-α, estimation as one of the death legend as its binding to its receptor results in the cleavage of Bid into a truncated form and causes Bax oligomerization and insertion into mitochon­dria, thus initiating apoptosis. (Table .4) .

As oxida­tive stress caused by overproduction of reactive oxygen species (ROS) is considered an apoptosis inducer. It is also estimated via measuring DNA fragmentation (Table 5). Figure 4A and B  .

10) Table 3: the results are quite surprising. Does mustard act on the cholesterol biosynthesis? Please discuss this point.

Replay:  Of course not, mustard doesn’t act on the cholesterol biosynthesis.  Our results are considered inside the normal range according to ( Smith and Mangkoewidjojo, 1998).  as he reported that blood cholesterol levels in wistar strain rats is ranging from [ 10-54 mg /dl] and this result is compatible with our result.

11)   The authors have to discuss the clinical translation of these finding

Reply: Although it is not literally clinical findings as we have done just an exploratory tests using experimental animals, but the findings were already discussed throughout the discussion part

Reviewer 2 Report

The manuscript by Hassan et al. describes the in vivo effect of plant mustard extracts on TAA-induced acute liver failure though a variety of inflammatory markers in mice serum as well as exploring their impact via genotoxic (comet assay) and immunohistological assessment. The biochemical part of the paper is clear and the results are interesting. However, I have few comments and suggestions, mainly on the synthetic part:

  1. In this study, the authors describe the synthesis of mustards-nano silver particles, not the synthesis of AgNPs, is it correct? Please revised this point in the manuscript (i.e. line 96, 108 etc)
  2. Line 102 and Figure 1: Why the color change indicates a complete transformation into mustard silver nanoparticles? The authors should better explain the encapsulation of mustard extracts into the particles and exclude any co-presence of AgNPs and extracts in the pellet after the centrifugation process.
  3. SEM and TEM (Figure 2 A and B) cannot be published in this state in the main draft, the quality should be improved. Also Figure 1 (x-axis) should be revised.
  4. Why the author selected nanoparticles process instead of liposomal technology?
  5. The title should be revised including the word "extracts".
  6. The connection between mustard extracts and TAA-induced acute liver failure is not clear in the text.
  7. Did the authors perform ROS assessment at least in selective liver culture cell lines to identify antioxidant effect of mustard extracts before working on mice?

Some minor changes/suggestions:

Line 18-19: ….. in male rats was  treated with LC/MS analysed mustard and its nanoform 100 g b.w before and after induction for 19 15 d.. , should be revises as …was treated with…… and analyzed with…

Line 20-21: the level of …. what?

Line 33: bio important organ, replace with important organ

Line 53, delete )

Line 68: anti-oxidative impact can be replaced with oxidative stress and anti-inflammatory impact

Line 101-102: Reduction of Ag to Ago. Please clear this point.

Line 105: 18 2.5. Clear these numbers?

Line 105-107: delete this sentence as the characterization is included at 2.5 section.

Line 180: replace "at room temperature"

Line 432 replace with "demonstrate"

Author Response

Dear Editors,:

Thank you very much for the review of our manuscript entitled: “Brassica juncea L(mustard) Nano Particles and Knocking off Oxidative Stress, Pro-Inflammatory Cytokine and Reverse DNA Genotoxicity “

Responses to the reviewer's comments

We would like to thank Reviewers for taking the time and effort necessary to review the manuscript. We sincerely appreciate all valuable comments and suggestions, which helped us to improve the quality of the manuscript. Our responses to the Reviewers’ comment are described below in a point-to-point manner. Appropriated changes, suggested by the Reviewers, have been introduced to the manuscript (highlighted within the document).

I would like to draw the reviewer attention that the line number has been changed due to the corrections done.

 (Reviewer 2):

Comments and Suggestions for Authors

The manuscript by Hassan et al. describes the in vivo effect of plant mustard extracts on TAA-induced acute liver failure though a variety of inflammatory markers in mice serum as well as exploring their impact via genotoxic (comet assay) and immunohistological assessment. The biochemical part of the paper is clear and the results are interesting. However, I have few comments and suggestions, mainly on the synthetic part:

  1. In this study, the authors describe the synthesis of mustards-nano silver particles, not the synthesis of AgNPs, is it correct? Please revised this point in the manuscript (i.e. line 96, 108 etc).

Reply: Thank you for this comment which we apologize for.  We already corrected and highlighted this sentence in the title and text.

  1. Line 102 and Figure 1: Why the color change indicates a complete transformation into mustard silver nanoparticles? The authors should better explain the encapsulation of mustard extracts into the particles and exclude any co-presence of AgNPs and extracts in the pellet after the centrifugation process.

Reply: Line 102 and Figure 1 already re-explained according to your suggestion as follow [Reduction of elemental Ag to Ago was confirmed by the color change from colorless to the brownish yellow indicating that mustard ethanolic extract was encapsulated into the silver nanoparticles in the form of pellets. The suspensions were subjected to centrifugation for the pellets to settle, which were then dried using a vacuum drier].     

  1. SEM and TEM (Figure 2 A and B) cannot be published in this state in the main draft, the quality should be improved. Also Figure 1 (x-axis) should be revised.

         Reply: - We done our best to get another photo with good quality of Figure2 A&B as much as we can   -Figure1 (x-axis) has been revised and marked with yellow color.

  1. Why the author selected nanoparticles process instead of liposomal technology?

Reply:  In our study we aimed to study the effect of nanoparticles as a simple easy technique. In addition nanoparticles have a benign nature and have an essential functional groups make AgNPs very stable and prevent nanoparticles from being anchored. In addition, the existence of the bioactive molecules on the surface of AgNPs may increase its efficacy and bioavailability .It can also enhance its solubility, improve plasma half-life, preventing degradation in the intestinal environment and increasing permeation in the small intestine which may confer strong anti oxidative activity.

  1. The title should be revised including the word "extracts".

Reply: The title is already revised.

  1. The connection between mustard extracts and TAA-induced acute liver failure is not clear in the text.

Reply:  Although the question is not fully understood, but we thought the connection is obviously demonstrated in the interaction between the oxidative stress induced by [TAA] and the rich antioxidant source [mustard extract] which improved the toxic effects of TAA.

  1. Did the authors perform ROS assessment at least in selective liver culture cell lines to identify antioxidant effect of mustard extracts before working on mice?

 Reply: previous study was done by ( Pandita et al.,2015) in vitro study.  But the main aim of this study is to estimate the effect of mustard in its two forms in vivo on the experimental animals.

Some minor changes/suggestions:

Line 18-19: ….. in male rats was  treated with LC/MS analyzed mustard and its nanoform 100 g b.w before and after induction for 19 15 d.. , should be revises as …was treated with…… and analyzed with

Reply:    Line 18-19: …was corrected to be. Mustard ethanolic extract was analyzed by HPLC/MS. To induce liver failure, male rats were injected with IP 350 mg / kg / b.w TAA,then treated with 100 mg b.w for 15 d of mustard extract and its nanoform before and following induction.

Line 20-21: the level of …. What?  

Reply:    Corrected  to be The levels of serum (AST), (ALT), (ALP), (TCHo), (TG), total bilirubin (TBIL), hepatic (MDA) and (NO), GSH, SOD, as well as TNF-α, and IL-6 were estimated.

Line 33: bio important organ, replace with important organ.   Reply:   Replaced already

Line 53, delete)         reply:     Deleted

Line 68: anti-oxidative impact can be replaced with oxidative stress and anti-inflammatory impact.                      Reply:      replaced with oxidative stress and anti-inflammatory impact.             

Line 101-102: Reduction of Ag to Ago. Please clear this point.  Reply:         Already cleared

Line 105: 18 2.5. Clear these numbers?              Reply:      Cleared

Line 105-107: delete this sentence as the characterization is included at 2.5 sections.

Reply:      Deleted.

Line 180: replace "at room temperature"           Reply:       replaced

Line 432 replace with "demonstrate"                  Reply:       replaced

Reviewer 3 Report

The study “Brassica juncea L (mustard) nano particles and Knocking off oxidative stress, pro-inflammatory cytokine and reverse DNA genotoxicity” investigates the role of a mustard extract and of mustard-extract-silver-nanoparticles in reducing thioacetamide induced hepatotoxicity in the rat.

A major problem with this study is that a proper control for silver-nanoparticles without mustard is missing. Therefore, it is not clear whether the effects are attributed to the silver or to mustard. Ag-nanoparticles might directly interact with TAA, as TAA can be chemically decomposed to H2S, which readily reacts with Ag+ to Ag2S. Thereby Ag-Nanoparticles might detoxify part of TAA in a chemical way. At least the reasons for this missing control and the implications should be properly discussed.

Also, is should be clearly stated everywhere that Ag-mustard-nanoparticles are studied (and not “mustard-nanoparticles”)!

The study tentatively identified some phenolic compounds in the mustard extract. I wonder why no glucosinolates were analyzed as mustard is very rich in these compounds and glucosinolate uptake and metabolism to isothiocyanates in the rat might also affect TAA toxicity…Using ethanol for extraction it is very likely that glucosinolates were extracted to. The study might benefit very much from this analysis.

Overall the presentation of the study in the current form is low in quality: The study is rich in typo errors and needs language editing. Moreover many references are mixed up/pint to the wrong references. All references need to be checked/corrected (It’s obvious in discussion part)! The methods part is incomplete and needs revision.

Moreover the following things needs to be revised:

Title: The title needs to be modified in order to make it clear that Ag-nanoparticles were studied.

Abstract: Avoid abbreviations or al lest explain them upon first mention! An abstract has to be a self-explanatory as it usually stands alone.

L20/21: the sentence is incomplete

L27: genotoxicity of what?

Keywords: add Ag-nanoparticles, avoid abbreviations (TAA is widely known abbreviation)

Introduction:

L51ff: Sentences incomplete, delete brackets

L65 and elsewhere: set “3” in AgNO3 to a lowercase letter

Materials and Methods:

L81f: What is meant by “sacked”? L82: How was evaporation done? The extract then was watery, without alcohol? How much extract was obtained from the 100g of mustard?

L85: Was the 100g mustard analyzed? I thought extract was analyzed? How was extract analyzed? L94: What is meant by reconstituted sample? Explain sample preparation!

L95: Add ESI settings, MS settings, how were compounds identified?

L96ff: How was the methanolic extract described here prepared? In section 2.2 an ethanolic extract is described.

L102: change to “…Ag+ to elemental Ag…)

L105/106: sentences incomplete

L119ff: How and after how many days of experiment were animals killed? How and which samples were taken? How were they handled/stored?

L144f: Explain in more detail? Which samples were analyzed? How were ALT; AST, ALP, TBIL analyzed?

Results:

Figure 1: X-axis need proper labelling (nanometer scaling), Ag-control in legend is red not green??? Figure captions need to state that Ag-nanoparticles are shown

Figure 2: needs to be prepared in better quality (can’t be read), Figure captions need to state that Ag-nanoparticles are shown

L194f: Explain compound identification. Were literature references used? If no reference compound was used for identification speak only of tentative identification!

L196ff (including Table 1): Here are loads of typos: I guess you mean “quinic” acid? Spaces are missing, caffeic needs to be corrected, “O” for oxygen always capital an in italic letters, correct “kaempferol”

Figure 3: Why methanolic?

Table 2-5 and throughout Results: Please always state that Ag-mustard-nanoparticles were studied, especially as “mustard” is only contained with 2% in this mixture!!!

L259ff: It seems that mustard and nanoparticles induce class 2 and 3 comets. So they seem to induce DNA-damage themselves. Please mention that finding and discuss it properly!

L279: Its unclear where this concentration originates from.

L338: Sentence is incomplete

L341: fatty acids were not analyzed in ethanolic extract.

Author Response

Dear Editors,:

Thank you very much for the review of our manuscript entitled: “Brassica juncea L(mustard) Nano Particles and Knocking off Oxidative Stress, Pro-Inflammatory Cytokine and Reverse DNA Genotoxicity “

Responses to the reviewer's comments

We would like to thank Reviewers for taking the time and effort necessary to review the manuscript. We sincerely appreciate all valuable comments and suggestions, which helped us to improve the quality of the manuscript. Our responses to the Reviewers’ comment are described below in a point-to-point manner. Appropriated changes, suggested by the Reviewers, have been introduced to the manuscript (highlighted within the document).

I would like to draw the reviewer attention that the line number has been changed due to the corrections done.

Reviewer 3:

Comments and Suggestions for Authors

The study “Brassica juncea L (mustard) nano particles and Knocking off oxidative stress, pro-inflammatory cytokine and reverse DNA genotoxicity” investigates the role of a mustard extract and of mustard-extract-silver-nanoparticles in reducing thioacetamide induced hepatotoxicity in the rat.

-A major problem with this study is that a proper control for silver-nanoparticles without mustard is missing. Therefore, it is not clear whether the effects are attributed to the silver or to mustard. Ag-nanoparticles might directly interact with TAA, as TAA can be chemically decomposed to H2S, which readily reacts with Ag+ to Ag2S. Thereby Ag-Nanoparticles might detoxify part of TAA in a chemical way. At least the reasons for this missing control and the implications should be properly discussed

Reply: previous studies (Wong et al.,2009) suggests that silver could have a therapeutic effect, here Ag-mustard-nanoparticles are studied .Pl note we have a control for mustard and another for Ag mustard nano particles.

-Also, is should be clearly stated everywhere that Ag-mustard-nanoparticles are studied (and not “mustard-nanoparticles”)

Reply:  we changed everywhere That Ag-mustard-nanoparticles were studied

-The study tentatively identified some phenolic compounds in the mustard extract. I wonder why no glucosinolates were analyzed as mustard is very rich in these compounds and glucosinolate uptake and metabolism to isothiocyanates in the rat might also affect TAA toxicity…Using ethanol for extraction it is very likely that glucosinolates were extracted to. The study might benefit very much from this analysis.

Reply: we would like to draw your attention that the extract is [ethanolic extract] done with ethanol or polar solvents using lC-MS to tentatively identify some phenolic compounds not the fatty acid, which needs nonpolar solvents .

-Overall the presentation of the study in the current form is low in quality: The study is rich in typo errors and needs language editing. Moreover many references are mixed up/pint to the wrong references. All references need to be checked/corrected (It’s obvious in discussion part)! The methods part is incomplete and needs revision.

Reply: The article is completely revised and corrected. References were also seriously checked

-Moreover the following things need to be revised:

-Title: The title needs to be modified in order to make it clear that Ag-nanoparticles were studied.   Reply: The title modified

-Abstract: Avoid abbreviations or al lest explain them upon first mention! An abstract has to be a self-explanatory as it usually stands alone. .

  Reply: the abstract revised to meet 200 words as required by the journal.

-L20/21: the sentence is incomplete  Reply: The sentence completed 

-L27: genotoxicity of what? Reply:    genotoxicity of DNA

-Keywords: add Ag-nanoparticles, avoid abbreviations (TAA is widely known abbreviation)

Reply:                           Ag-nanoparticles has been added

-Introduction:

L51ff: Sentences incomplete, delete brackets.                         Reply:  Deleted

L65 and elsewhere: set “3” in AgNO3 to a lowercase letter.

 Reply:                         already modified to lower case elsewhere

-Materials and Methods:

L81f: What is meant by “sacked”? L82: How was evaporation done? The extract then was watery, without alcohol? How much extract was obtained from the 100g of mustard?  Reply:  sorry for writing mistake .It was modified to be [Then left for 72 h. for about 3days].

-L82: How was evaporation done? Reply:  evaporation was done by rotary evaporation [Added already].

-How much extract was obtained from the 100g of mustard?  Reply:  (9.8 gm was obtained from the 100g of mustard [already added to this part]

-L85: Was the 100g mustard analyzed? I thought extract was analyzed? How was extract analyzed? L94: What is meant by reconstituted L85: sample? Explain sample preparation![

Reply:  The yield obtained out of the 100g mustard was analyzed   The analysis was done in the ethanolic extract of the yield 12.5gm and it was analyzed by LC/MS.

-Explain sample preparation?

Reply: The sample preparation was described in details in section 2.2.

-What is meant by reconstituted L85: sample?

Reply: This part was completely modified according to reviewer recommendation Section 2.3. Titled [LC/MS analysis for ethanolic Extract of mustarad B. juncea L]. 

-L95: Add ESI settings, MS settings, how were compounds identified?

Reply: ESI settings, MS settings already added Section. 2.3

-L96ff: How was the methanolic extract described here prepared? In section 2.2 an ethanolic extract is described.

 Reply:  It is ethanolic not methanolic but methanol is used as mobile phase only.

-L102: change to “…Ag+ to elemental Ag…) Reply:  changed to elemental Ag

-L105/106: sentences incomplete :) Reply   Corrected &completed

-L119ff: How and after how many days of experiment were animals killed? How and which samples were taken? How were they handled/stored?  

Reply:  So Sorry This part was completely added and highlighted in yellow color.

L144f: Explain in more detail? Which samples were analyzed? How were ALT; AST, ALP, TBIL analyzed

Reply:  Explained with more details in section 2.7[ Biochemical analysis.]

Results:

-Figure 1: X-axis need proper labelling (nanometer scaling), Ag-control in legend is red not green??? Figure captions need to state that Ag-nanoparticles are shown

Reply:  Figure 1 is completely modified proper labeling was written . Ag-control in legend is red. Figure captions started with Ag-nanoparticles

-Figure 2: needs to be prepared in better quality (can’t be read), Figure captions need to state that Ag-nanoparticles are shown .

Reply:  prepared in better quality &the caption corrected

-L194f: Explain compound identification. Were literature references used? If no reference compound was used for identification speak only of tentative identification!

Reply: The compound identifications part  was added and inserted to section 3[Result ]

-L196ff (including Table 1): Here are loads of typos: I guess you mean “quinic” acid? Spaces are missing, caffeic needs to be corrected, “O” for oxygen always capital an in italic letters, correct “kaempferol”

Reply: has been checked and corrected.

-Figure 3: Why methanolic? Reply:   Corrected.

Table 2-5 and throughout Results: Please always state that Ag-mustard-nanoparticles were studied, especially as “mustard” is only contained with 2% in this mixture!!!

Reply:   Corrected to Ag-mustard-nanoparticles in table 2-5 and throughout results

-L259ff: It seems that mustard and nanoparticles induce class 2 and 3 comets. So they seem to induce DNA-damage themselves. Please mention that finding and discuss it properly!

Reply:    Yes it looks like this .

For this we tried to re -study this part again .unfortunately there was no remaining freezed liver tissue. in addition to the difficult conditions and more restrictions settled down  currently due to corona virus, we couldn’t  offer the same previous  conditions to repeat that test again. Moreover there is a possibility that the seeds itself have traces of pesticides which affected the DNA as a sensitive structure. But the exact mechanism is not fully -clear which may need further studies. So we built our interpretation only with regard to the various treated groups on the obvious difference between the -ve and Positive control. Further we depending more on the histo and immunohistopathological results, which confirmed the improvement illustrated with mustard in comparison to TAA positive control.

-L279: It’s unclear where this concentration originates from.

Reply: 100mg/kg is dose conc of mustard extract  given in I ml volume to the investigated rats

-L338: Sentence is incomplete     

Reply: Has been completed

-L341: fatty acids were not analyzed in ethanolic extract.

Reply: we would like to draw your attention that the extract is [ethanolic extract] done with ethanol or polar solvents using lC-MS to tentatively identify some phenolic compounds not the fatty acid, which needs nonpolar solvents.

Discussion Line 472-475   Recently many various antioxidants were shown to rip the liver damage. Within the cutting-edge look at we located significant increase in the levels of each of GSH and, SOD when treated with mustard extract and its nano form, whether pre or post TAA injection

Has been changed to [It is well established that plant extracts with antioxidant potential can protect against the oxidative damage caused by TAA hepatotoxicity [2]. The present study confirmed this observation and demonstrated that mustard extract and its nano form, substantially increased the levels of GSH and, SOD whether pre or post TAA injection.] 

Round 2

Reviewer 1 Report

The revised version is much more better than the first version.

However, the authors did not correct the paper accordingly to my comments.

The main issue is the wrong statistical method used in the paper. I suggested in my report to the authors to use a two-way ANOVA instead of of one way ANOVA. I highly suggest the authors to change it.

Author Response

Dear editor

Dear reviewers,

Thank you for your email and sincerely appreciate the constructive comments concerning our Manuscript ID: biomolecules-976197, entitled “Brassica juncea L (mustard) nano particles and Knocking off oxidative stress, pro-inflammatory cytokine and reverse DNA genotoxicity .

We have carefully studied the comments and Our responses to the Reviewers’ comment are described below in a point-to-point manner. Appropriated changes, suggested by the Reviewers, have been introduced to the manuscript (highlighted within the document). we hope it will meet your kind approval.

Reviewer 1:

The revised version is much better than the first version.

However, the authors did not correct the paper accordingly to my comments.

The main issue is the wrong statistical method used in the paper. I suggested in my report to the authors to use a two-way ANOVA

Response:   Kindly be informed that the statistics was done based on the comparison of the studied groups with one variable at a time eg: comparison of the studied groups with ALT in one time and re compared again  with AST,ALP in another time and so on with each individual marker measured throughout the study eg table below in blue color. After comparing all measured parameters  individually with the studied groups we gathering them in one table like Table  2, 3, 4.5 

studied Groups

ALT (U/mL)

Mean± SD (Improvement %0

Control (-ve)

53b± 7.4

TAA (+ve)

124a± 10.20

Control-Mustard (Ms)

51.4b ± 8.45

Control- Ag nano Mustard (NMs)

62.8 b± 7.89

Prophylactic-Mustard (PMs)

68b ± 8.60 (105%)

Prophylatic Ag-nano-Mustard (PNMs)

62.2b± 9.20 (116%)

Treated-Mustard (TMs)

58.2 b ± 8.50 (125%)

Treated Ag nano-Mustard (TNMs)

55.4 b ± 11.09 (130%)

The following are some similar references with similar studies following the same way of analysis for your kind perusal.

1-Ali SA, Ibrahim NA, Mohammed MM, El-Hawary S, Refaat EA. The potential chemo preventive effect of ursolic acid isolated from Paulownia tomentosa, against N-diethylnitrosamine: initiated and promoted hepatocarcinogenesis. Heliyon. 2019 May 1;5(5):e01769.

Data presented as mean _ standard error (SE), Statistical analysis is carried out using one way analysis of variance (ANOVA) Computer program by Co Stat Computer Program significant at p _ 0.05.

2- Motawi TK, Ahmed SA, El-Boghdady NA, Metwally NS, Nasr NN. Impact of betanin against paracetamol and diclofenac induced hepato-renal damage in rats. Biomarkers. 2020 Jan 2;25(1):86-93. Results were expressed as mean± SEM and statistical comparison were carried out using one-way analysis of variance (ANOVA),

3-Yimam, M., Jiao, P., Moore, B., Hong, M., Cleveland, S., Chu, M., ... & Kim, M. R. (2016). Hepatoprotective activity of herbal composition SAL, a standardize blend comprised of Schisandra chinensis, Artemisia capillaris, and Aloe barbadensis. Journal of nutrition and metabolism2016.‏ Results were expressed as mean± SEM and statistical comparison were carried out using one-way analysis of variance (ANOVA),

Reviewer 2 Report

The authors addressed my comments point-by-point and my suggestions have been included in the manuscript.

Author Response

Thank you very much for your guidance

Reviewer 3 Report

The manuscript improved in some ways. However many of my comments unfortunately have not been taken seriously (which is frustrating), therefore the article still needs revision:

1) In response letter Authors state:previous studies (Wong et al.,2009) suggests that silver could have a therapeutic effect, here Ag-mustard-nanoparticles are studied .Pl note we have a control for mustard and another for Ag mustard nano particles.”

Where is the control for mustard and another for Ag mustard nano particles? A control for Ag mustard nanoparticles would be AG nanoparticles alone. Can’t find this!

2) Old comment “The study tentatively identified some phenolic compounds in the mustard extract. I wonder why no glucosinolates were analyzed as mustard is very rich in these compounds and glucosinolate uptake and metabolism to isothiocyanates in the rat might also affect TAA toxicity…Using ethanol for extraction it is very likely that glucosinolates were extracted to. The study might benefit very much from this analysis.

Reply: we would like to draw your attention that the extract is [ethanolic extract] done with ethanol or polar solvents using lC-MS to tentatively identify some phenolic compounds not the fatty acid, which needs nonpolar solvents .”

-> The response makes no sense. I asked whether glucosinolates were not analysed (not for fatty acid). Glucosinolates are polar compounds and ARE EXTRACED with ethanol.

3) -Abstract: AGAIN My comment was not taken seriously: The abstract still is full of unexplained abbreviations, which is not acceptable as no one will aunderstand the abstract as a stand-alone. Please avoid abbreviations in the abstract or at least explain them upon first mention! If the abstract is too long you have to shorten it!

4) “Old comment-L95: Add ESI settings, MS settings, how were compounds identified?

ReplyESI settings, MS settings already added Section. 2.3”

->> No they are NOT given in 2.3!: Give all ESI and MS settings (temperatures, voltages, gas flows. Scan range, …)

5) Please speak of “tentative” identification throughout the manuscript. Only those compounds can be termed identified whose identity was verified with a reference compound or NMR spectrum!

6) Old comment: -L259ff: It seems that mustard and nanoparticles induce class 2 and 3 comets. So they seem to induce DNA-damage themselves. Please mention that finding and discuss it properly!

Reply:    Yes it looks like this .

For this we tried to re -study this part again .unfortunately there was no remaining freezed liver tissue. in addition to the difficult conditions and more restrictions settled down  currently due to corona virus, we couldn’t  offer the same previous  conditions to repeat that test again. Moreover there is a possibility that the seeds itself have traces of pesticides which affected the DNA as a sensitive structure. But the exact mechanism is not fully -clear which may need further studies. So we built our interpretation only with regard to the various treated groups on the obvious difference between the -ve and Positive control. Further we depending more on the histo and immunohistopathological results, which confirmed the improvement illustrated with mustard in comparison to TAA positive control.

-> Why do you not discuss/mention the finding (It seems that mustard and nanoparticles induce class 2 and 3 comets.)?????? It’s a result of your study, so it should be also discussed and taken seriously.

 Others:

L89 and elsewhere: “gm” is no unit; please change to “g”.

L105: AGAIN: How was the “1mL” extract prepared? In 2.2. a dry residue is described (12g dried extract).

L269 The Table 1 has still many mistakes!!! Correct “Caffieic acid hexoside” -> Caffeic acid hexoside; “Kaemperol-O-cafeoyldihexoside”-> Kaempferol-O-caffeoyldihexoside; “O” for oxygen always capital an in italic letters; decide to use capital letters or not, but don’t make a mixture

- The study is still rich in typo errors and missing spaces (or to much spaces) and needs language editing. 

Author Response

Dear Editor:

Dear reviewers,

Thank you for your efforts and sincerely appreciate the constructive commentsand value guidance concerning our Manuscript ID: biomolecules-976197, entitled “Brassica juncea L (mustard) nano particles and Knocking off oxidative stress, pro-inflammatory cytokine and reverse DNA genotoxicity .

We have carefully studied the comments and our responses to the Reviewers’ comment are described below in a point-to-point manner. Appropriated changes, suggested by the Reviewers, have been introduced to the manuscript (highlighted within the document). We hope it will meet your kind approval.

Reviewer 3 :

The manuscript improved in some ways. However many of my comments unfortunately have not been taken seriously (which is frustrating), therefore the article still needs revision:

1) In response letter Authors state: “previous studies (Wong et al.,2009) suggests that silver could have a therapeutic effect, here Ag-mustard-nanoparticles are studied .Pl note we have a control for mustard and another for Ag mustard nano particles.” Where is the control for mustard and another for Ag mustard nano particles? A control for Ag mustard nanoparticles would be AG nanoparticles alone. Can’t find this!

The authors sincerely appreciate your time and the constructive comments which we done our best to answer it thoroughly as much as we can.

Response1:

 For your kind attention this study was designed to investigate The protective effects mustard and of silver nanoparticles synthesized by green chemistry using mustard.   So we aimed to synthesis [mustards-nano silver particles, not to study or synthesis of AgNPs .

In another words mustard extract was used as a reducing and stabilizing agent for AgNo3. So mustard acts as a capping reduced agent only to transform mustard particles into nano scales, but not to investigate the effect of the silver or the pure nano silver itself as it was previously widely studied. Pl find some similar studies synthesized sivernano particles using green plants without any Ag nano control.

  • Zhang Z, Xin G, Zhou G, Li Q, Veeraraghavan VP, Krishna Mohan S, Wang D, Liu F. Green synthesis of silver nanoparticles from Alpinia officinarum mitigates cisplatin-induced nephrotoxicity via down-regulating apoptotic pathway in rats. Artificial cells, nanomedicine, and biotechnology. 2019 Dec 4;47(1):3212-21. 2-    
  • Sengottaiyan, A., Aravinthan, A., Sudhakar, C., Selvam, K., Srinivasan, P., Govarthanan, M., ... & Selvankumar, T. (2016). Synthesis and characterization of Solanum nigrum-mediated silver nanoparticles and its protective effect on alloxan-induced diabetic rats. Journal of Nanostructure in Chemistry6(1), 41-48
  • Dobrucka R, Szymanski M, Przekop R. The study of toxicity effects of biosynthesized silver nanoparticles using Veronica officinalis extract. International Journal of Environmental Science and Technology. 2019 Dec; 16(12):8517-26.

2) Old comment “The study tentatively identified some phenolic compounds in the mustard extract. I wonder why no glucosinolates were analyzed as mustard is very rich in these compounds and glucosinolate uptake and metabolism to isothiocyanates in the rat might also affect TAA toxicity…Using ethanol for extraction it is very likely that glucosinolates were extracted to. The study might benefit very much from this analysis.

Reply: we would like to draw your attention that the extract is [ethanolic extract] done with ethanol or polar solvents using lC-MS to tentatively identify some phenolic compounds not the fatty acid, which needs nonpolar solvents .”

-Reviewer> The response makes no sense. I asked whether glucosinolates were not analysed (not for fatty acid). Glucosinolates are polar compounds and ARE EXTRACED with ethanol

Response2:

Yes although mustard is very rich with glucosinolate but it is not analyzed. We would like to draw your attention that we were interested and focus more to identify the phenolic compounds to study their synergetic anti oxidative properties in our study.

For co authors suggestions) -Abstract: AGAIN My comment was not taken seriously: The abstract still is full of unexplained abbreviations, which is not acceptable as no one will aunderstand the abstract as a stand-alone. Please avoid abbreviations in the abstract or at least explain them upon first mention! If the abstract is too long you have to shorten it!

Response2:

We apologize for that. Although we were very careful to be adhered with the journal restrictions as the Abstract should be only 200 word. The abstract was re modified according to your comment with complete explained abbreviations, so everyone can understand. [highlighted with green color].

4) “Old comment-L95: Add ESI settings, MS settings, how were compounds identified?

Reviewer Reply: ESI settings, MS settings already added Section. 2.3”

->> No they are NOT given in 2.3!: Give all ESI and MS settings (temperatures, voltages, gas flows. Scan range, …)

Response 4: ESi settings, MS settings temperatures, voltages, gas flows. Scan range already added Section, 2.3”highlighted” in green color.

5) Please speak of “tentative” identification throughout the manuscript. Only those compounds can be termed identified whose identity was verified with a reference compound or NMR spectrum

Response 5:  “tentative “was added throughout the manuscript.

6) Old comment: -L259ff: It seems that mustard and nanoparticles induce class 2 and 3 comets. So they seem to induce DNA-damage themselves. Please mention that finding and discuss it properly!

Reply:    Yes it looks like this .

For this we tried to re -study this part again .unfortunately there was no remaining freezed liver tissue. in addition to the difficult conditions and more restrictions settled down  currently due to corona virus, we couldn’t  offer the same previous  conditions to repeat that test again. Moreover there is a possibility that the seeds itself have traces of pesticides which affected the DNA as a sensitive structure. But the exact mechanism is not fully -clear which may need further studies. So we built our interpretation only with regard to the various treated groups on the obvious difference between the -ve and Positive control. Further we depending more on the histo and immunohistopathological results, which confirmed the improvement illustrated with mustard in comparison to TAA positive control.

Reviewer -> Why do you not discuss/mention the finding (It seems that mustard and nanoparticles induce class 2 and 3 comets.)?????? It’s a result of your study, so it should be also discussed and taken seriously.

Response 6: Thank you very much for your value notice. The findings were discussed and highlighted in green in the discussion part.

 Others:

L89 and elsewhere: “gm” is no unit; please change to “g”.

Response:  gm has been corrected to [g] highlighted in green

L105: AGAIN: How was the “1mL” extract prepared? In 2.2. a dry residue is described (12g dried extract).

Response:  Depending on the previous calculated mustard concentration100mg/kg body weight. The yield (12g dried extract) was dissolved in distilled water to be given to the rats in a dose of 1ml [means that 1ml contains 100mg/kg body weight]. This part illustrated in section 2.6.1. Experimental protocol line 150 highlighted in green.

.

L269 tthe Table 1 has still many mistakes!!! Correct “Caffieic acid hexoside” -> Caffeic acid hexoside; “Kaemperol-O-cafeoyldihexoside”-> Kaempferol-O-caffeoyldihexoside; “O” for oxygen always capital an in italic letters; decide to use capital letters or not, but don’t make a mixture

Response:  Table 1 has been corrected & highlighted in green

- The study is still rich in typo errors and missing spaces (or to much spaces) and needs language editing. 

Response: Typo errors and missing spaces corrected and complete reviewing has been done to the manuscript highlighted in in yellow & green .